# High-Accuracy Neural Network Interatomic Potential for Silicon Nitride

**DOI:** 10.3390/nano13081352

**Published:** 2023-04-13

**Authors:** Hui Xu, Zeyuan Li, Zhaofu Zhang, Sheng Liu, Shengnan Shen, Yuzheng Guo

**Affiliations:** 1The Institute of Technological Sciences, Wuhan University, Wuhan 430072, China; 2School of Power and Mechanical Engineering, Wuhan University, Wuhan 430072, China; 3School of Electrical and Automation, Wuhan University, Wuhan 430072, China

**Keywords:** molecular dynamics, machine learning, amorphous silicon nitride, density functional theory, deep potential

## Abstract

In the field of machine learning (ML) and data science, it is meaningful to use the advantages of ML to create reliable interatomic potentials. Deep potential molecular dynamics (DEEPMD) are one of the most useful methods to create interatomic potentials. Among ceramic materials, amorphous silicon nitride (SiN_x_) features good electrical insulation, abrasion resistance, and mechanical strength, which is widely applied in industries. In our work, a neural network potential (NNP) for SiN_x_ was created based on DEEPMD, and the NNP is confirmed to be applicable to the SiN_x_ model. The tensile tests were simulated to compare the mechanical properties of SiN_x_ with different compositions based on the molecular dynamic method coupled with NNP. Among these SiN_x_, Si_3_N_4_ has the largest elastic modulus (*E*) and yield stress (*σ*_s_), showing the desired mechanical strength owing to the largest coordination numbers (CN) and radial distribution function (RDF). The RDFs and CNs decrease with the increase of *x*; meanwhile, *E* and *σ*_s_ of SiN_x_ decrease when the proportion of Si increases. It can be concluded that the ratio of nitrogen to silicon can reflect the RDFs and CNs in micro level and macro mechanical properties of SiN_x_ to a large extent.

## 1. Introduction

Molecular dynamics (MD) mainly rely on Newtonian mechanics to simulate the motion of molecular systems. Compared with the expensive cost of the experiment, the MD method is a cost-effective and efficient tool for exploring the properties of various complicated new materials. To make sure the simulation results match well with the experiments, it is important to use an accurate description of interatomic interactions. Quantum mechanics (QM) simulations, such as the ab initio molecular dynamics (AIMD) method based on the density functional theory (DFT), are the most reliable way to describe the atomic interactions for different systems [1,2,3,4,5]. Although AIMD exhibits desirable computational accuracy, the time cost of AIMD is very high, which limits the application of AIMD. To balance the calculation performance and speed of MD simulations, empirical interatomic potentials have been applied to MD simulations. Empirical interatomic potentials, including Lennard–Jones (LJ) [6], embedded atom method (EAM) [7,8], the Stillinger–Weber (SW) [9,10], Tersoff [11] and charge-optimized many-body (COMB) [12] potentials obviously improved the calculation performance and speed of MD. However, the application of these empirical potentials is hindered by their poor transferability. In terms of systems described by two-body interactions, the LJ potential has favorable accuracy. The SW and Tersoff potentials are able to combine two-body and three-body interactions to stabilize tetrahedral solids, but the description accuracy of bond breaking and metallic phases of silicon and carbon is not sufficient. Owing to the rapid development of machine learning (ML) and data science, it is meaningful to take the advantages of ML to create reliable ML interatomic potentials to replace the conventional empirical potentials [13,14,15]. In 2007, J. Behler and M. Parrinello first proposed a method to create ML interatomic potentials based on artificial neural network deep learning [16]. Various ML potentials have been generated and further applied to material property calculations. Among these ML potential training methods, deep potential molecular dynamics (DEEPMD) is one of the most useful methods to create interatomic potentials recently [16,17]. Results prove that the various systems that used DEEPMD methods delivered good accuracy [18,19]. For example, Wang et al. developed a DEEPMD potential to describe the properties of Li-Si alloys [20]. The DEEPMD potential is about 20 times faster than the AIMD simulations. Meanwhile, the accuracy of DEEPMD potential is comparable with that of AIMD.

Amorphous silicon nitride (SiN_x_) has been widely used in advanced semiconductor packaging owing to its good electrical insulation, abrasion resistance, and mechanical strength [21,22,23].

However, compared to the plenty of experimental studies about SiN_x_, there is less research about SiN_x_ on theoretical calculations at present [24,25,26]. In this work, we conducted a preliminary and comprehensive theoretical calculation of SiN_x_. At the same time, considering that current classic MD potentials are not suitable for SiN_x_ simulations due to efficiency and accuracy [27,28,29,30,31,32], it is of great interest to train a new potential to describe the interatomic interactions.

In our work, a neural network potential (NNP) was trained based on the ML in the DEEMD package for SiN_x_ with the 3:4 composition, owing to its high stability. To validate the accuracy of the NNP, the energies and forces obtained by the NNP were compared with its AIMD counterpart. The structure and properties of Si_3_N_4_ calculated -by MD + NNP were compared with the AIMD through the radial distribution functions (RDFs), coordination numbers (CNs), and bond angle distributions (BADs). Similar to the empirical potentials, the NNP trained by DEEPMD can also be used for SiN_x_ with different compositions. Therefore, the NNP was confirmed to be reliable and applicable for SiN_x_. In the following parts, SiN_x_ all refers to the amorphous silicon nitride with different compositions. Then, the NNP was applied to MD simulations to predict the properties of SiN_x_. To further investigate the influence of compositions on the mechanical properties of SiN_x_, the tensile tests of SiN_x_ were carried out at 300 K. Among these SiN_x_ models, the Si_3_N_4_ has the desired mechanical strength, which is consistent with the microstructure, including the maximal RDF peak and CNs.

## 2. Materials and Methods

DEEPMD is one of the most popular methods to create interatomic potentials recently owing to its calculation speed and accuracy [17,33]. The details of the NNP training process are listed as follows.

### 2.1. AIMD Calculations

In our work, Si_3_N_4_ was chosen as an example to train the NNP because of the high stability of the 3:4 composition. The training database of Si_3_N_4_ at different temperatures was obtained by first-principles calculations using the Vienna ab initio simulation package (VASP) [34]. The exchange–correlation interaction was described by Perdew–Burke–Ernzerhof (PBE) functional [35]. The interaction between electrons and ions was described by the projector-augmented wave (PAW) approach. The cut-off energy is 520 eV. We use the k-point mesh grid with a spacing of 0.4 Å within the Gamma-centered k-sampling to sample the Brillouin zone. The initial Si_3_N_4_ configuration with 112 atoms is a cube with randomly distributed atoms; the model was built by LAMMPS “create_atoms” command and modified by Material Studio. The periodic boundary conditions were applied in *x*, *y*, and *z* directions. The cube size in the *x*, *y,* and *z* directions is 11.935 Å, 11.935 Å, and 11.55 Å, respectively. The AIMD calculations were fulfilled at a constant temperature of 2000 K with an NVT ensemble. The Nose-Hoover thermostat was used to control the temperature of the AIMD simulation. The timestep is 1 fs running 10,000 steps. The energy and force errors less than 10^−5^ eV/atom and 0.01 eV/Å, respectively, are convergence criteria for geometry optimization.

### 2.2. Deep Potential Training Process

Based on the AIMD calculations, 10,000 data points were transferred from the output file “OUTCAR”; the data points were divided into five sets. The four sets were used for training databases, with the remaining one selected as a testing database. The smooth edition of the DEEPMD and DeepPot-SE model, implemented in the DEEPMD-kit package [33], was used to train the interatomic potential. The cutoff radius of the model is 6.0 Å for neighbor searching with the smoothing function starting from 5.8 Å. The hidden layers were divided into three layers, and the number of neurons in each layer is 25, 50, and 100, respectively. The learning rate starts from 0.001 using a decay rate of 0.95 every 5000 steps. The decay step and stop learning rate are 5000 and 3.51 × 10^−8^, respectively. Based on the datasets of AIMD calculations at 2000 K, the NNP potential at 2000 K was obtained. In order to obtain a high-quality potential, a training database including different temperatures is crucial; as a result, the same method was used for the 3000 K potential training as well. Then, the 2000 K and 3000 K NNP potentials were combined to achieve a new NNP. Finally, the NNP was frozen, and the frozen model can be used in model testing and MD simulations.

## 3. Results and Discussions

### 3.1. The Accuracy of NNP

To verify the accuracy of NNP at 2000 K and 3000 K, the energies and forces from NNP after DEEPMD training were compared with those from AIMD shown in Figure 1 and Figure 2. The results show that the validation data generally distribute around the *y* = *x* line, showing that the obtained potential can predict the energies and forces precisely. The root-mean-square errors (RMSEs) and R-Square (R^2^) of energies were calculated to evaluate the performance of NNP. The smaller RMSEs mean better prediction. On the contrary, the higher R^2^, between 0 and 1, indicates better results. It can be seen that the NNP with larger R^2^ and smaller RMSE at *y* and *z* directions, respectively, works better than that of the *x* direction, both in Figure 1 and Figure 2. Meanwhile, the R^2^ and RMSE are acceptable for NNP at *x* direction as well.

To further validate the reliability of NNP, the RDFs, CNs, and BADs of Si_3_N_4_ calculated by NNP were compared with those of AIMD. To research these properties of Si_3_N_4_, a simulation at 2000 K was conducted by a large-scale atomic/molecular massively parallel simulator (LAMMPS) [36] with the MD [37]. The detailed simulation parameters are shown in the Appendix A. The RDFs, also known as pair correlation function, usually refer to the distribution probability of other particles in the Δ*r* thickness shell at the distance *r* of a specified particle. RDFs are widely used to study the degree of order of materials and describe the correlation of atoms, which can be calculated by
(1)$$g_{\alpha \beta }(r)=\frac{V}{N_{\alpha }N_{\beta }}\sum_{\alpha }\frac{\langle N_{\alpha \beta }(r,△r)\rangle }{4\pi r^{2}}$$
where *V* is the volume of the simulation cell. *N_α_* and *N_β_* are the number of *α*-type ions and *β*-type ions, respectively. *N_αβ_* (*r*, Δ*r*) is the average number of *α*-type ions around *β*-type ions in a spherical space. The results of the NNP + MD and AIMD simulations in Figure 3a confirm the accuracy of the NNP. The MD in the following content is specified as MD coupled with NNP. It can be seen that the first peak values of Si_3_N_4_ RDFs calculated by MD are located around *r* = 1.7 Å. The results show that the distribution function *g*(*r*) obtained by MD is consistent with the AIMD results calculated by VASP. The same conclusion can be reached at 3000 K in Figure 4a. Therefore, it is concluded that the NNP is capable of predicting structure information of SiN_x_ with AIMD accuracy within this range from 2000 K to 3000 K.

CNs are the number of coordination atoms around the central atom of a compound. The RDFs depend on the multilayered coordination radius and coordination particle numbers. The CNs could be calculated by integration of the RDFs
(2)$$N=4\pi \rho \int_{0}^{R_{\min}}g_{\alpha \beta }(r)r^{2}dr$$
where *N* is the CN. *ρ* is particle density in the simulation cell. *R*_min_ is the first minimum in Figure 3a and Figure 4a. As is shown in Figure 3b, we now concentrate on the CNs of Si_3_N_4_. The results calculated by MD are compared with the AIMD simulation results. The CNs calculated by MD fit the AIMD results accurately, as well as in Figure 4b. The CNs are proportional to the pair distance and CNs are zero when the pair distance is less than 1.5 Å, so the minimal pair distance of Si_3_N_4_ is 1.5 Å. The BADs were calculated to analyze the local geometries of the first coordination shell, which can be calculated by
(3)$$\theta_{ijk}=\langle \cos^{-1}\frac{r_{ij}^{2}+r_{ik}^{2}-r_{jk}^{2}}{2r_{ij}r_{ik}}\rangle$$
where *θ* is the bond angle. *r*_ij_, *r*_ik,_ and *r*_jk_ are the bond length between atoms. The BADs of Si_3_N_4_ obtained by MD and AIMD at 2000 K and 3000 K are shown in Figure 5 and Figure 6, respectively. At 2000 K, there are few BAD smaller than 45°. The majority of BADs are localized between 75° and 120°, and the Si-N-Si and N-Si-N peak values of MD BADs are 90°, which is consistent with AIMD BADs. As for 3000 K, though the curves are rougher than that of 2000 K, the Si-N-Si and N-Si-N peak values of MD BADs are consistent well with AIMD BADs. Therefore, NNP is reliable enough to predict CNs and BADs information of SiN_x_. Based on the RDFs, CNs, and BADs analysis above, it can be concluded that we can use the NNP to calculate the structure information of SiN_x_.

### 3.2. The NPP Applied for Structure Information

Similar to the empirical potentials, the NNP can also be used for different compositions with the same elements. To study the effect of compositions on SiN_x_ structures, we used the NNP to simulate the heating process of SiN_x_ and analyzed the simulation results, including RDFs, CNs, and BADs. The models of SiN_x_ were built by LAMMPS, as shown in Figure 7. The parameters of SiN_x_ models are shown in Table 1. The boundary condition, timestep, ensemble, and other parameters of MD simulations are the same as part 3 for Si_3_N_4_.

The RDFs of SiN_x_ at 2000 K and 3000 K are shown in Figure 8. In terms of a given temperature, the *g*(*r*) peak values decrease while the proportion of Si composition increase, indicating that the pair distance decrease in SiN_x_ with a higher Si proportion. In the case of 2000 K, the peak and valley values fluctuate more obviously. As the temperature increases to 3000 K, there are few differences between the peak values of different compositions, indicating that all the SiN_x_ are in the same phase and there is less difference in microstructure as the temperature increases. The heating process for SiN_x_ was calculated as well with the RDFs at different temperatures shown in Appendix A.

During the heating process, CNs and BADs at 3000 K are shown in Figure 9. The CNs of SiN_x_ in Figure 9a share the same profile, which is all proportional to the pair distance. Among these CNs, SiN_x,_ with a 3:4 composition, has the largest CN, which corresponds to the largest *g*(r) in the Si-poor model. Therefore, Si_3_N_4_ is expected to deliver the most stable configuration at the micro level. With the increase of x in SiN_x_, the CNs decrease slightly, indicating that the CNs are weakly affected by composition. The BADs of SiN_x_ at 3000 K are shown in Figure 9b,c. Similar to the CNs, the composition has little influence on the BADs, especially for the N-Si-N BADs in Figure 9c. In summary, the RDFs, CNs, and BADs of SiN_x_ are different, so the structures of SiN_x_ are different as well. In the next section, we used tensile simulations to explain the impact of different compositions on SiN_x_ mechanical properties from a macro perspective.

### 3.3. The NNP Applied for Tensile Tests

To evaluate the accuracy of tensile by MD, the simulated strain–stress curve of Si_3_N_4_ compared with the experimental result is shown in Figure 10. The elastic modulus *E* of the simulation and experiment is 284.6 GPa and 253.3 GPa, respectively, which are consistent well with the reported *E* in the previous reference [38,39]. As is known, the *E* depends on the size of the model [39,40]; the simulated *E* 10.9% higher than the experimental case is acceptable for the tensile test. It is impossible to fabricate the perfect single crystal Si_3_N_4_ in the experiment. The Si_3_N_4_ mechanical strength is determined by the defects and grain boundaries, so the experimental mechanical strength of Si_3_N_4_ is typically smaller than those of calculated results. The tensile tests of Si_3_N_4_ at different temperatures were calculated as well shown in Appendix A. The results show that the elastic modulus and peak values vary inversely with the temperature increasing, so when the temperature is 300 K, the mechanical properties of Si_3_N_4_ perform better than those at 1000 K and 3000 K. Since the accuracy of MD has been confirmed, the tensile tests of SiN_x_ with compositions from 3:4 to 1:1 were calculated at 300 K. The tensile models of SiN_x_ were repeated to supercells from the stable structures in part 4 with parameters of tensile models shown in Appendix A. The cross-sections of SiN_x_ in the tensile tests are shown in Figure 11. It can be seen that the cross-section of Si_3_N_4_ is flat, and the strain is small, so the Si_3_N_4_ cracked immediately and exhibited brittleness properties. In terms of other SiN_x_, the lengths in the *y* direction are larger than that of Si_3_N_4_ when the fractures occurred, indicating that it takes a long time for the SiN_x_ to crack. With the increase of *x*, the SiN_x_ starts to exhibit flexibility, especially the SiN with the largest deformation when the fracture occurred.

The strain–stress curves and mechanical properties of SiN_x_ are shown in Figure 12. All the curves have a linear increase at the initial stage, which corresponds to an elastic deformation. The slopes of SiN_x_ at the initial linear stage are different, indicating that the *E* varies with different compositions. The *E* is an important parameter of materials at the macro level, which represents the ability of an object to resist elastic deformation and reflects the bond strength between atoms, ions, or molecules at the micro level. The Si_3_N_4_ curve has a steep slope and large yield stress *σ*_s_ in the linear stage, illustrating that Si_3_N_4_ is a typical brittle material. On the contrary, another SiN_x_ demonstrates ductile property, especially the SiN with an obvious yield stage. When the strain ranges from 10% to 22%, the SiN curve becomes flat in the yield stage. The flexibility of SiN significantly improved compared with its Si_3_N_4_ counterpart, which is consistent with the analysis in Figure 8. The *E* of the maximal and minimal slope is 349.78 GPa and 138.39 GPa for *x* = 4/3 and *x* = 1/1, respectively. The *E* is 199.68 GPa, 188.95 GPa, 240.82 GPa, and 160.48 GPa, respectively, when *x* = 5/4, 6/5, 7/6, and 8/7. Besides the *E*, the *σ*_s_ of SiN_x_ are different as well, showing that fracture occurs at different stages during the tensile simulations. The maximal *σ*_s_ is 27.45 GPa of Si_3_N_4_. Although the *σ*_s_ of other SiN_x_ decrease with higher x, the *σ*_s_ of SiN_x_ fail to vary inversely with the *x* due to the amorphous structures of SiN_x_ (*x* = 5/4, 6/5, 7/6, and 8/7). Among these curves, the 3:4 composition has the maximal *E* and *σ*_s_, which can correspond to the largest RDFs and CNs in Figure 8 and Figure 9, respectively. The RDFs and CNs decrease with the decreasing of x, leading to the flexibility improvement of SiN_x_, since for Si_3_N_4_ *x* = 4/3 while for SiN *x* = 1. Meanwhile, the *E* of SiN_x_ excluding Si_3_N_4_ decrease compared with that of Si_3_N_4_, which is generally inversely proportional to the RDFs. The results show that the RDFs and CNs at the micro level can reflect the macro mechanical properties of SiN_x_ to a large extent.

## 4. Conclusions

In this work, the interatom potential for SiN_x_ was created by the DEEPMD kit with a neural network. Based on the comparison of energies and forces from NNP and AIMD for tranSi_3_N_4_, we find that NNP can easily achieve DFT accuracy. The RDFs, CNs, and BADs simulations between the NNP and AIMD confirmed that the NNP is reliable enough to calculate the structure information Si_3_N_4_ as well. Then, we conducted comprehensive calculations to predict the RDFs, CNs, and BADs of SiN_x_. Therefore, we used tensile simulations to explain the impact of different compositions on SiN_x_ properties from a macro perspective. Si_3_N_4_ is a typical brittle material with the largest *E* and *σ*_s_. The flexibility of SiN_x,_ excluding Si_3_N_4,_ improved, leading to the decrease of *E* and *σ*_s_. The *E* and *σ*_s_ fail to be inversely proportional with *x* due to the amorphous structures of SiN_x_. The RDFs and CNs at the micro level can reflect the macro mechanical properties of SiN_x_ to a large extent. Among these compositions, *x* = 4/3 features high mechanical strength, owing to the largest CN and RDF. The *E* of SiN_x,_ excluding Si_3_N_4,_ decreases compared with that of Si_3_N_4_, leading to the flexibility improvement of SiN_x_, which is generally inversely proportional to the RDFs.

## Figures and Tables

**Figure 1 nanomaterials-13-01352-f001:**
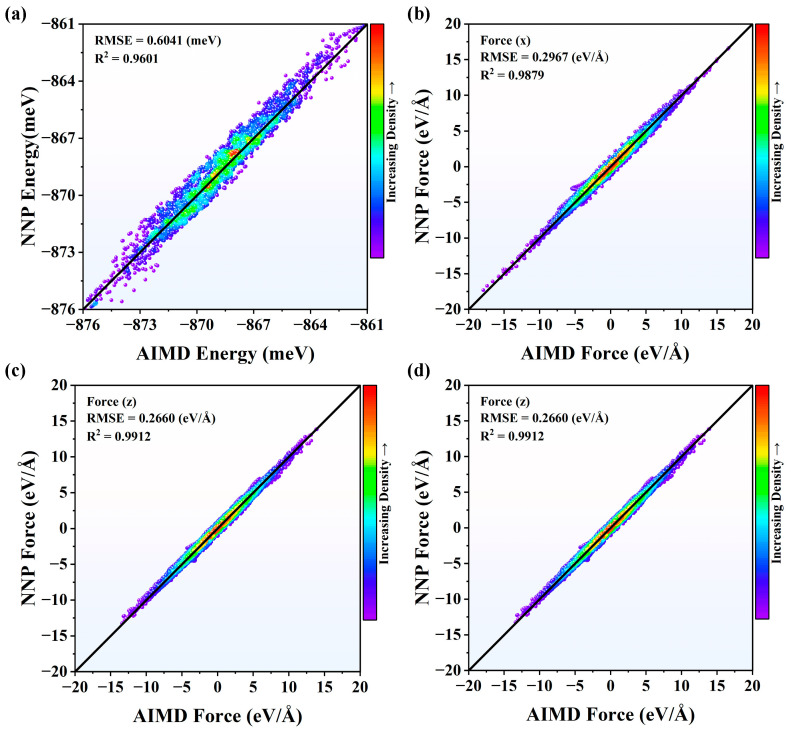
The comparisons of (**a**) energies and (**b**–**d**) forces from NNP and AIMD for Si_3_N_4_ at 2000 K in *x*, *y*, and *z* directions.

**Figure 2 nanomaterials-13-01352-f002:**
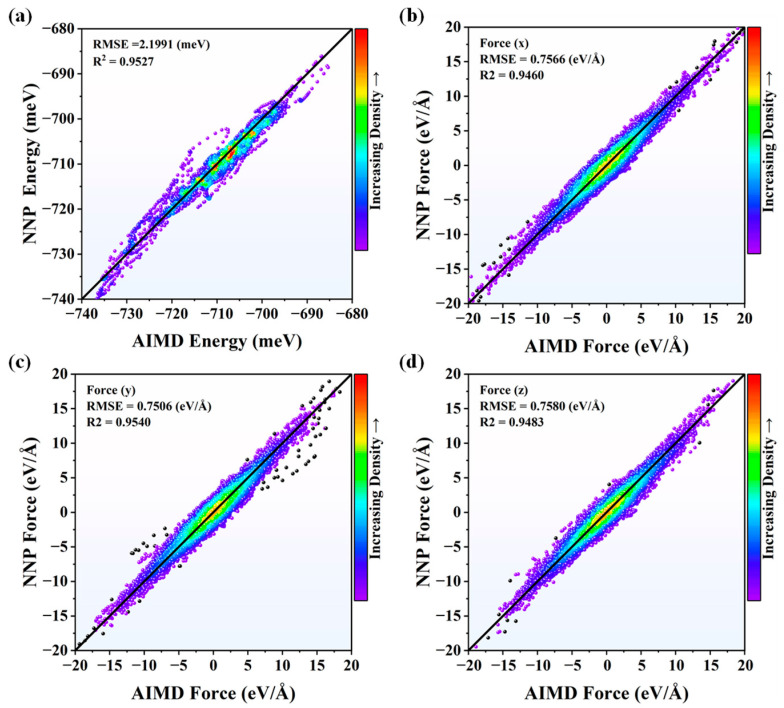
The comparisons of (**a**) energies and (**b**–**d**) forces from NNP and AIMD for Si_3_N_4_ at 3000 K in *x*, *y*, and *z* directions.

**Figure 3 nanomaterials-13-01352-f003:**
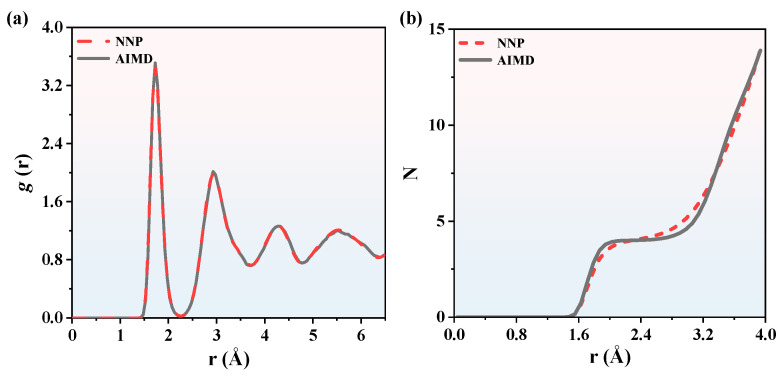
The comparisons of (**a**) Si_3_N_4_ RDFs between NNP + MD and AIMD calculations and (**b**) Si_3_N_4_ CNs between NNP + MD and AIMD calculations. The temperature is 2000 K.

**Figure 4 nanomaterials-13-01352-f004:**
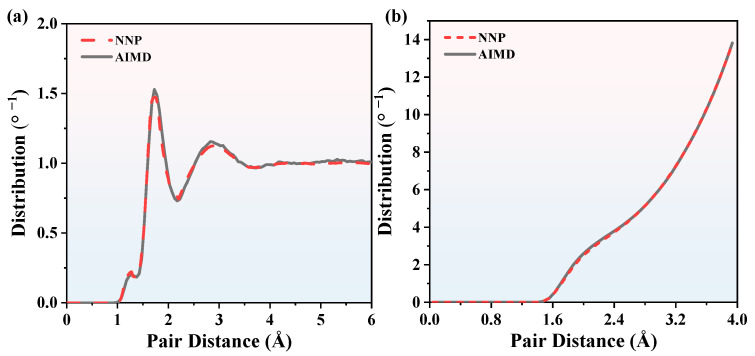
The comparisons of (**a**) Si_3_N_4_ RDFs between NNP + MD and AIMD calculations and (**b**) Si_3_N_4_ CNs between NNP + MD and AIMD calculations. The temperature is 3000 K.

**Figure 5 nanomaterials-13-01352-f005:**
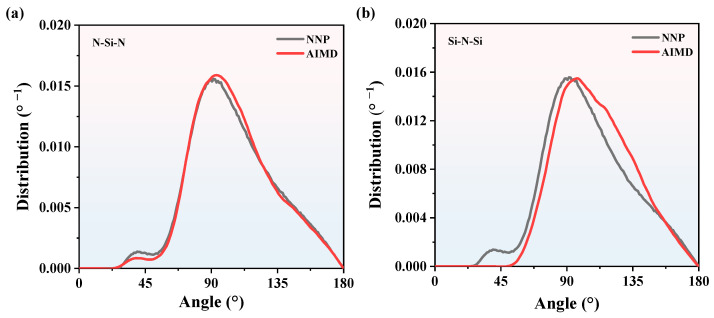
The BADs comparison of Si_3_N_4_ between NNP + MD and AIMD at 2000 K. (**a**) the N-Si-N BADs, (**b**) the Si-N-Si BADs.

**Figure 6 nanomaterials-13-01352-f006:**
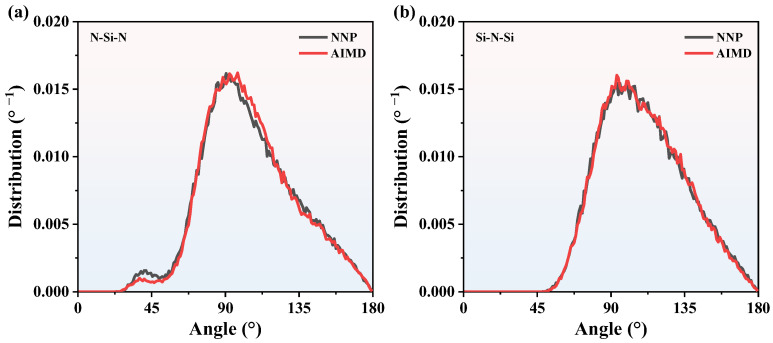
The BADs comparison of Si_3_N_4_ between NNP + MD and AIMD at 3000 K. (**a**) the N-Si-N BADs, (**b**) the Si-N-Si BADs.

**Figure 7 nanomaterials-13-01352-f007:**
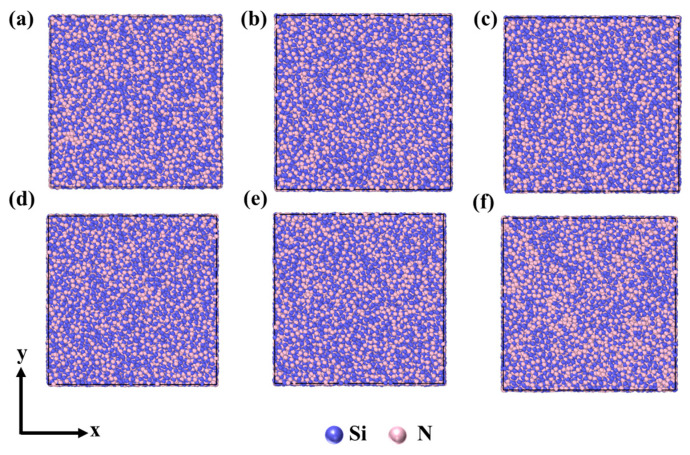
The models of SiN_x_. (**a**) *x* = 4/3, (**b**) *x* =5/4, (**c**) *x* = 6/5, (**d**) *x* = 7/6, (**e**) *x* = 8/7, (**f**) *x* = 1/1.

**Figure 8 nanomaterials-13-01352-f008:**
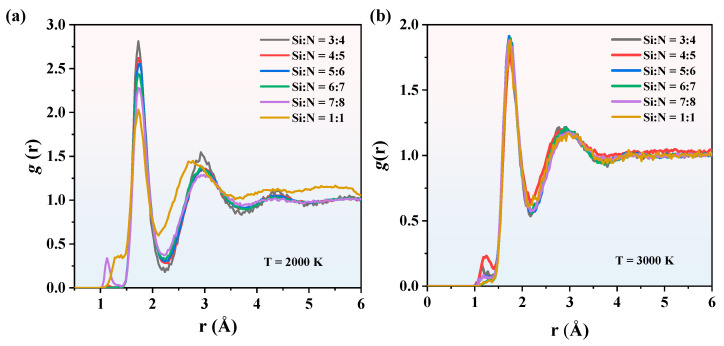
The RDFs of SiN_x_ at (**a**) 2000 K and (**b**) 3000 K.

**Figure 9 nanomaterials-13-01352-f009:**
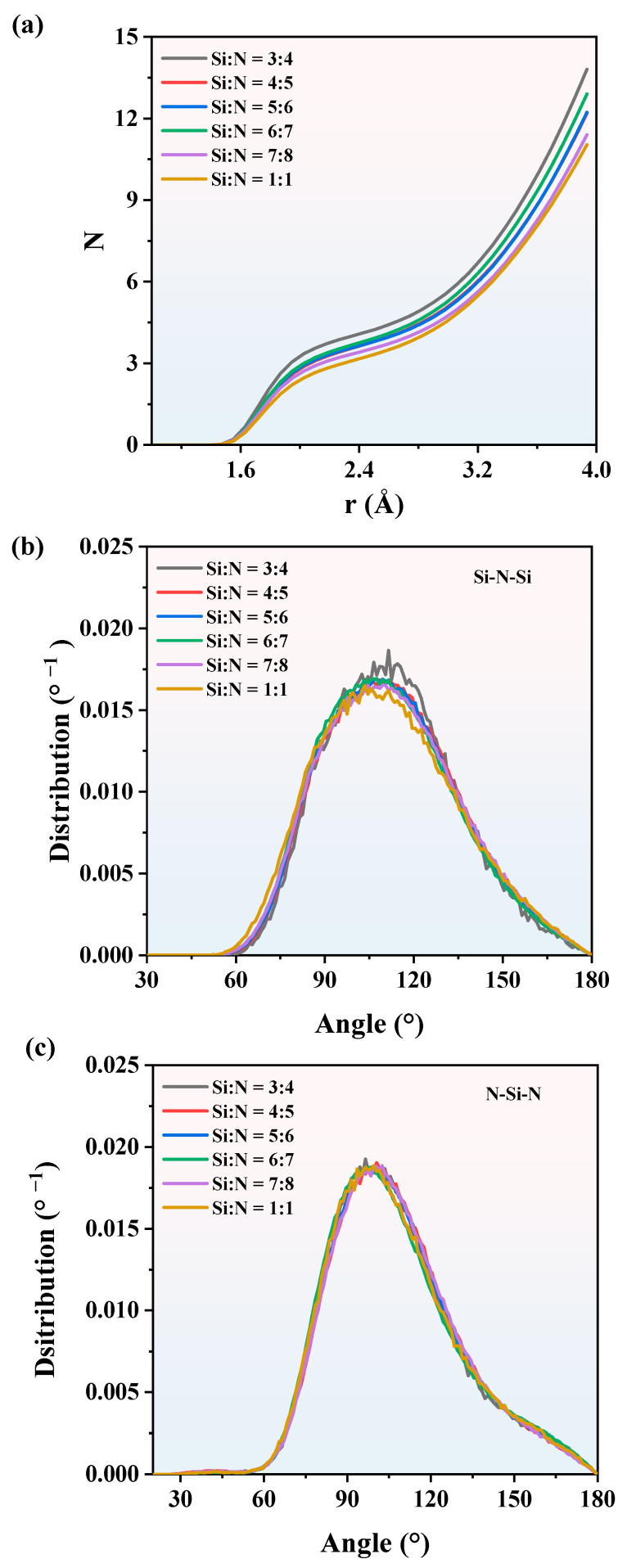
(**a**) The CNs of SiN_x_ at 3000 K. The BADs of (**b**) Si-N-Si and (**c**) N-Si-N at 3000 K.

**Figure 10 nanomaterials-13-01352-f010:**
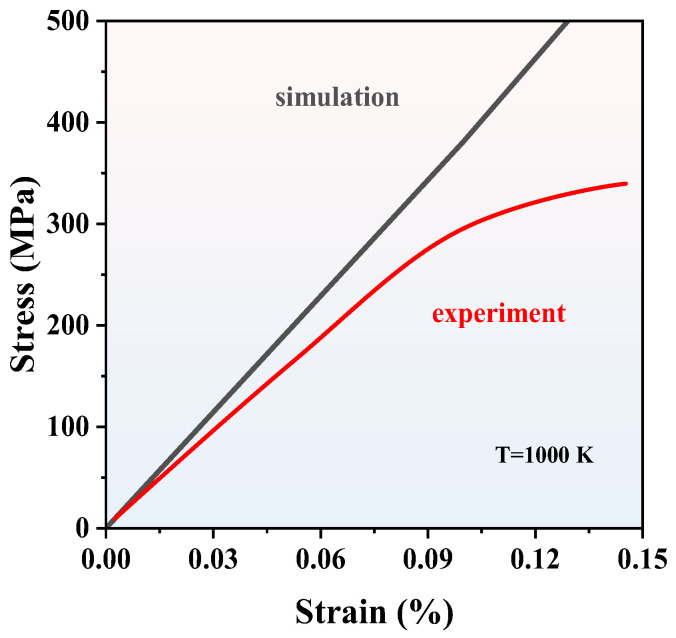
The simulated and experimental [39] strain–stress comparison of Si_3_N_4_. Reproduced with permission [39]. Copyright 2018 Elsevier.

**Figure 11 nanomaterials-13-01352-f011:**
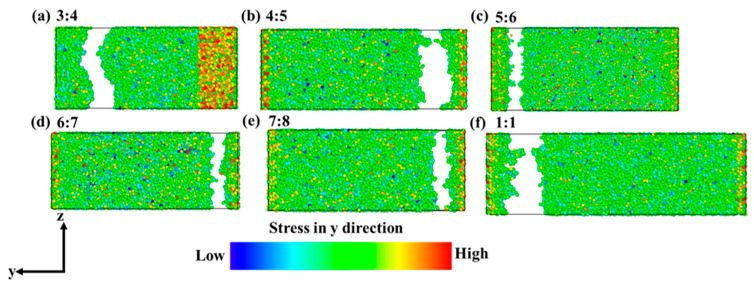
The cross-sections of SiN_x_ in the tensile tests. (**a**) *x* = 4/3, (**b**) *x* =5/4, (**c**) *x* = 6/5, (**d**) *x* = 7/6, (**e**) *x* = 8/7, (**f**) *x* = 1/1.

**Figure 12 nanomaterials-13-01352-f012:**
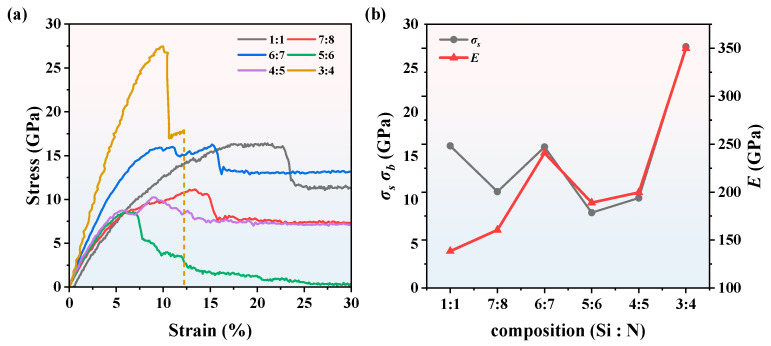
(**a**) The strain–stress curves of SiN_x_ at 300 K. (**b**) The mechanical properties of SiN_x_.

**Table 1 nanomaterials-13-01352-t001:** The simulation parameters in SiN_x_ heating process.

Si:N	*x* (Å)	*y* (Å)	*z* (Å)	Atoms
4:5	54	54	54	13500
5:6	54	54	54	13750
6:7	54	54	54	14625
7:8	54	54	54	13125
1:1	54	54	54	13500

## Data Availability

The data that support the findings of this study are available from the corresponding author upon reasonable request.

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
