# Peer review of "High-Accuracy Neural Network Interatomic Potential for Silicon Nitride"

_nanomaterials, 2023, doi:10.3390/nano13081352_

Round 1

Reviewer 1 Report

Xu et al. reported a theoretical/computational study of the development and application of a molecular force field to simulate silicon nitride (SiNx) and investigate this nanomaterial’s ideal composition for optimal mechanical properties. This work used deep learning to train the silicon nitride force field parameters and validated that it has comparable accuracy as in the ab initio simulation results. Then the authors also investigated other Si:N compositions and found that Si3N4 has the largest elastic modulus and acts as a typical brittle material.

The main theme and most results of this manuscript in general meet the scope and standard of Nanomaterials. Therefore, I recommend its acceptance, after some minor concerns (detailed below) are adequately addressed in a revised version.

Minor Concerns:

1. In section 2.2, the authors briefly described the training process of this “deep potential” for silicon nitride systems. However, a lot of key information for building and training the molecular model is still missing. What is the formula for this DEEPMD atomic potential? What is the exact “data” (Coordinates? Energy? Force?) used for training? What are the final parameters after training? This information might already be included in Ref [17] and [33], but it would still be necessary to elaborate in this manuscript.

2. Most of the AIMD and DEEPMD performed in this work are at extremely high temperatures (2000 K or 3000 K). What is the temperature range for silicon nitride to behave as a useful nanomaterial? Are the temperatures used in the simulations within this range? 

3. In Figure 2, the authors calculated the radial distribution function (RDF) and coordination number (CN) of the system. However, since the simulated systems are binary mixtures of silicon and nitride, there are three possible types of RDF and CN in total (Si-Si, Si-N, N-N). Which type of RDF and CN were calculated in the results here? 

4. In Figure 8, why did the fractures of the material occur in the x direction, while the stress was applied in the y direction? 

Reviewer 2 Report

The authors report machine learned potentials for SiN. In general, this is an interesting paper, but there are major questions the authors need to address

"poor transferrability problem" needs more explanations. Transferrability  from what to what? TraPPE is also a transferrable forcefield, for example. Sort of. (I am not insisting the authors describe TraPPE, I am saying that deficiencies of the existing models have to be specified better).

Referencing: I insist the authors compare their approach to MLIP (Shapeev et al), since MLIP is probably #1 ML approach to crystal simulations (I am not a co-author of any of MLIP papers, just it is very strange that the authors disregards MLIP)

The accuracy issue: the authors compare energies and forces, but i fact there are no "absolute energies" and therefore the RMSE is in fact meaningless. The authors need some metrics there, a phase transition heat for SiN or something like that. In terms of forces: pressure vs density, that should be easy to calculate. Otherwise F1 has no real outcome

181 As the temperature increases to 3000 K, there is few differences between the peak values of different compositions, indicating that SiNx phase transition occurs at 3000 K. -- Which transition and how this corresponds to the exp behaviour?

Figure 5: the peak (no matter how minor ) at very short distances should be a major concern, but are not even mentioned. What kind of configurations are behind them and are they physical?

The authors break SiN creating a surface. ML potentials trained on bulk phases are known for poor performance when interfaces are modeled. Could the authors comment on that, since they extensively discuss the crack shape?

Finally , journal suitability: the authors model a bulk phase with a 3D PBCs, which is no more "nanostructured" than crystalline quartz. Why "Nanomaterials"?

editorial:

some sentences are hard to understand

213 the mechanical strength of Si3N4 at 300 K outperforms that of different temperature

219 In terms of other SiNx, the crossections are rougher and have adhesions indicating that it takes a process for the SiNx to crack. With the increasing of x, the SiNx strat to exhibit flexibility, especially the SiN with the largest deformation when fracture happened.

179 peak values vary inversely with the composition of Si, indicating that the pair distance decrease in Si-poor composition.

Reviewer 3 Report

The authors created a neural network interatomic potential (NNP) for silicon nitride, and tested the accuracy of NNP for Si3N4. They conducted systematical calculations of RDFs, CNS, BADs, and E for SiNx systems. The work is well organized and presented. There are several minor issues listed below that need to be addressed before considering publication in this journal. 

1. In line 136, "NNP+MD" better "MD using NNP". 

2. In line 176, for Si3N4, if writing in SiNx format, x=4/3, not 3:4. Please correct x values for all SiNx systems accordingly. 

3. In line 247, "RDFs and CNs decrease with the increasing of x" should be "RDFs and CNs decrease with the decreasing of x" since for Si3N4 x=4/3 while for SiN x=1. 

Round 2

Reviewer 2 Report

I am almost fine with the responses.

One point that remains are energy RMSE. Again: there are no "absolute energies" as the authors write in the response. Forces are physically meaningful as well as the energy differences, but not energies on their own. A simple example is a dihedral angle: if I include 1-4 vdW and electrostatic interactions in the energy function (eg CHARMM forcefield) I can still fit the energy profile for the angle less some constant. The behaviour of the system obtained in MD|MC would be practically the same. However, if one fits another forcefield to the energies calculated with AMBER and CHARMM forcefields, RMSE would be different for fits of the same actual accuracy. There must be some baseline with which energies are compared. Some forcefields actually include that baseline, but that should be explained as well.

With reference to MLIP: I did not require the authors to actually try the MLIP server (that would have been way too much for a review), I just think thy should compare their approach to deriving the machine-learned potentials for solid system to the existing approaches: authors should justify why they do not simply follow existing paths. 
